

# Structural analysis of urinary light chains and proteomic analysis of hyaline tubular casts in light chain associated kidney disorders

Thomas Reiter[1], Daniela Knafl[1], Hermine Agis[2], Karl Mechtler[3], Ludwig Wagner[1] and Wolfgang Winnicki[1]

[1] Department of Medicine III, Division of Nephrology and Dialysis, Medical University of Vienna, Vienna, Austria
[2] Department of Medicine I, Division of Oncology, Medical University of Vienna, Vienna, Austria
[3] ProtChem Facility, IMP-IMBA, Research Institute of Molecular Pathology, Vienna, Austria

## ABSTRACT

**Background**. Monoclonal overproduction of kappa and/or lambda light chains might result in renal light chain deposition disease. Light chain associated cast nephropathy and renal AL-amyloidosis represent two further pathologies going along with monoclonal gammopathy of renal significance and multiple myeloma. While cast nephropathy often manifests with acute kidney injury, AL-amyloidosis is rather accompanied with chronic kidney disease.

**Methods**. Urine samples were collected from 17 patients with multiple myeloma or monoclonal gammopathy. The urine sediment was stained for cast morphology by H/E and light chain immunofluorescence. Following micro-selection of casts under microscope, proteomic analysis of casts was performed by mass spectrometry. Sucrose gradient sedimentation was employed and light chain architecture examined by immunoblotting. Uromodulin was measured by ELISA in sucrose gradient fractions.

**Results**. Urinary casts were observed of about 30 μm in diameter by H/E staining and under immunofluorescence microscopy. Casts with a diameter of 20 μm were observed as a novel variant. Proteome analysis showed that in addition to the expected light chain variants produced by the malignant clone of plasma cells, also histones such as H2B and cathepsin B were contained. Uromodulin was not detectable in urinary casts of all patients. All eleven patients with lambda light chains showed predominant dimerized light chains in the urine immunoblot. Six patients with kappa light chains presented with predominantly monomeric forms of light chains in the immunoblot. The densitometric evaluated ratio of lambda dimers vs. monomers was significantly higher ($2.12 \pm 0.75$) when compared with the ratio of kappa dimers vs. monomers ($0.64 \pm 0.47$), $p = 0.00001$. Aggregates of light chains separated in part into denser sucrose fractions.

**Conclusion**. This work on urinary casts and light chains demonstrates that hyaline tubular casts represent a complex formation of protein-protein aggregates with histones and cathepsin B identified as novel cast components. Apart from the proteomic composition of the casts, also the formation of the light chains and aggregates is of relevance. Dimerized light chains, which are typical for lambda paraproteins, might

Corresponding author
Ludwig Wagner,
ludwig.wagner@meduniwien.ac.at

be less dialyzable than monomeric forms and may therefore identify patients less responsive to high cut-off dialysis.

## INTRODUCTION

Renal manifestation of multiple myeloma or elevated free light chains due to monoclonal gammopathy frequently manifest in light chain associated kidney disorders with different forms of deposits in renal compartments (*Leung et al., 2019*). The spectrum of renal complications associated with paraproteins is heterogeneous, with two major categories corresponding to the properties of the underlying B-cell clone. The first group is always associated with a high tumor mass and in this context cast nephropathy is of particular significance. The second group includes all patients with renal symptoms associated with a nephrotoxic monoclonal immunoglobulin as clonal proliferative disorder and is defined as monoclonal gammopathy of renal significance (MGRS) (*Leung et al., 2012*). Depending on results of immunofluorescence and electron microscopy, MGRS-associated diseases may include immunoglobulin light chain (AL) amyloidosis, monoclonal fibrillary glomerulonephritis (GN), immunotactoid GN, cryoglobulinaemic GN, light-chain proximal tubulopathy, crystal storing histiocytosis, monoclonal immunoglobulin deposition disease as well as proliferative glomerulonephritis and monoclonal immunoglobulin deposits (*Leung et al., 2019*; *Leung et al., 2012*).

Monoclonal overproduction of light chains by plasma cells represents the origin of the renal pathology. Since the publication of these studies nephrologists and oncologists have learned much in this direction (*Hutchison et al., 2011*; *Manohar, Nasr & Leung, 2018*). Diagnostic procedures have been extended to include measurement of free light chain ratio in the serum, protein electrophoresis in serum and urine, immunofixation in serum and urine, bone marrow and renal biopsy (*Leung, 2016*), whereas free light chain measurement in the urine is not recommended (*Bradwell et al., 2003*; *Dispenzieri et al., 2009*; *Leung et al., 2019*). Urine sediment testing for presence of urinary casts is simple, noninvasive and not harmful to patients and is included in any diagnostic workup for patients with acute kidney injury (AKI).

Tubular cast formation has been attributed to specific non-covalent binding of light chains with uromodulin proteins (*Huang & Sanders, 1997*). This is provoked when the proximal tubular epithelia are overloaded with high concentrations of kappa or lambda light chains which under normal conditions are reabsorbed by proximal tubular epithelial cells through a multiligand endocytic receptor complex (*Ying et al., 2012*). This receptor complex is assumed to be constituted out of megalin (*Klassen et al., 2005*) and cubilin (*Batuman et al., 1998*) as silencing of both genes resulted in ablation of free light chain induced uptake and toxicity in human renal proximal tubule epithelial cells (*Ying et al., 2012*). The primary structure of the monoclonal free light chain which results in a defined

isoelectric point of the protein has much influence on the morphology and location of the light chain disease (*Abraham et al., 2003*) such as cast nephropathy (*Cogne et al., 1991*; *Stevens, 2000*), AL-amyloidosis (*Enqvist et al., 2007*; *Omtvedt et al., 2000*; *Stevens, 2000*) or monoclonal immunoglobulin deposition disease (*Del Pozo Yauner et al., 2008*; *Deret et al., 1997*). However, individual patient-specific predisposing factors and medication (*Holland et al., 1985*; *Ying & Sanders, 1998*) might represent similar important players in renal disease morphology and progression. In this respect, secretion and reabsorption of other proteins and pharmacotherapeutic compounds (*Luque et al., 2017*) in urine appear to have a significant effect on cast formation. There is currently evidence that the human urinary proteome consists of more than 1000 individual protein fragments shaded or secreted into the urine (*Liu et al., 2012*; *Marimuthu et al., 2011*). It has already been reported that renal casts have variable hematoxylin/eosin staining intensities and might contain further compounds beyond uromodulin and light chain (*Luque et al., 2017*).

These previous studies have motivated us to investigate the protein constituents of nephron obstructing tubular casts through specific staining and micro-selection of these casts in the urinary sediment. Casts are usually observed in the distal tubule, however, there is evidence that they are also found in the proximal tubule and even in glomeruli (*Start et al., 1988*). In this study analyzing urinary secreted casts, there are indications that thin casts might even originate from the loop of Henle. Proteome analysis of these casts might help to shed further light on genesis and potential treatment strategies in the clinical work up of patients with deranged free light chain composition in urine. Free light chain dimerization or multimerization may influence clearance properties when high cut-off membrane dialysis is intended. Therefore, in addition to staining of urine sediment and proteomic analysis of casts, immunoblotting of urinary light chains was an important focus of this study.

## MATERIALS & METHODS

### Urine sediment

Morning urine samples were collected from 17 patients with multiple myeloma or monoclonal gammopathy of renal/undetermined significance. Demographics, clinical course of the patients and their predominant paraprotein are depicted in Table 1. After centrifugation of the urine at 3,000 g for 10 min, the supernatant was frozen at $-80\,°C$ for later use in fluid analysis. The pellet was re-suspended in 2.5 mL cell culture media (RPMI 1640 containing 10% calf serum) and 100 μL were applied to the funnel of a Shandon cyto-centrifuge. Following a spin at 1200 RPM for 3 min cytoslides were air dried for at least two hours and then either immediately processed as described below or wrapped in aluminum foil and frozen at $-25\,°C$ for further analysis.

**Hematoxylin/Eosin staining** was carried out as used in routine hematology staining of blood smears. Slides were covered in Depex mounting media and a cover slip was applied for microscopic observation.

**Immunofluorescence for anti kappa/lambda staining** (confocal microscopy) was carried out after fixation in acetone for 4 min. The cell and cast containing area was

Reiter et al. (2019), *PeerJ*, DOI 10.7717/peerj.7819

Peer J

**Table 1** Demographics, clinical characteristics and specification of light chain associated kidney disorder.

| ID | Age (years) | Gender | LC | Hematological classification | Renal histology | Disease duration (years) | sCr[b] (mg/dL) | Urinary cast | CKD/AKI | U:P/C (mg/g) | Previous therapy | Illustration |
|---|---|---|---|---|---|---|---|---|---|---|---|---|
| 1 | 69 | f | λ | MM | LCPT | 1 | 1.24 | 0 | 1 | 112 | 1,3 | Figs. 4 and 5/lane1 |
| 2 | 76 | f | λ | MM | nd | 10 | 0.81 | 0 | 1 | 81 | 5,6 | Figs. 4 and 5/lane2 |
| 3 | 76 | m | λ | MGRS | LCPT | 4 | 1.2 | 0 | 4 | 1,714 | | Figs. 4 and 5/lane3 |
| 4 | 87 | f | κ | MM | nd | 6 | 1.07 | 0 | 2 | 1 | 8 | Figs. 4 and 5/lane4 |
| 5 | 63 | f | κ | MM | nd | 6 | 0.61 | 0 | 1 | 274 | 1,3,4 | Figs. 4 and 5/lane5 |
| 6 | 49 | m | κ | MM[a] | nd | 1 | 1.12 | 0 | 2 | 214 | 1,4,7 | Figs. 4 and 5/lane6 |
| 7 | 75 | m | κ | MM | nd | 0 | 2.12 | 0 | 5 | 697 | 3 | Figs. 4 and 5/lane7 |
| 8 | 57 | f | λ | MM[a] | LCCN | 5 | 5.33 | 0 | 5 | 767 | 1,2,4 | Figs. 4 and 5/lane8 |
| 9 | 62 | m | λ | MGUS | nd | 0 | 0.84 | 0 | 4 | 476 | | Figs. 4 and 5/lane9 |
| 10 | 63 | m | λ | MGUS | nd | 0 | 1.05 | 0 | 4 | 273 | | Figs. 4 and 5/lane10 |
| 11 | 67 | f | λ | MM[a] | TMA | 4 | 2.02 | 0 | 4 | 222 | 1,4 | Figs. 4 and 5/lane11 |
| 12 | 80 | m | κ | MM | nd | 0 | 1.04 | 0 | 2 | 126 | 2,4,5 | Figs. 4 and 5/lane12 |
| 13 | 67 | f | λ | MM[a] | LCCN, LCPT, AL-amyloidosis | 1 | 1.59 | pos | 5 | 5,631 | 1,7 | Figs. 4 and 5/lane13; Fig. 6 |
| 14 | 55 | m | λ | MM | LCCN | 5 | 1.22 | pos | 4/3 | 4,781 | 1 | Figs. 4 and 5/lane14; Fig. 3; Sup. Fig 1 |
| 15 | 59 | m | κ | MM | LCCN | 1 | 6.58 | pos | 3/3 | 5,561 | 1,2 | Fig. 7 |

*(continued on next page)*
**Table 1** (*continued*)

| ID | Age (years) | Gender | LC | Hematological classification | Renal histology | Disease duration (years) | sCr[b] (mg/dL) | Urinary cast | CKD/AKI | U:P/C (mg/g) | Previous therapy | Illustration |
|---|---|---|---|---|---|---|---|---|---|---|---|---|
| 16 | 75 | m | λ | MM[a] | LCCN, AL-amyloidosis | 0 | 4.2 | pos | 5 | 4,643 | | ns |
| 17 | 48 | m | λ | MM[a] | nd | 7 | 0.92 | 0 | 1 | 38 | 1,2,4 | ns |

**Notes.**

AKI, acute kidney injury; CKD, chronic kidney disease; f, female; ID, identification; LC, light chain; LCCN, Light-chain cast nephropathy; LCPT, light-chain proximal tubulopathy; m, male; MGRS, monoclonal gammopathy of renal significance; MGUS, monoclonal gammopathy of undetermined significance; MM, multiple myeloma; nd, not done; ns, not shown; sCr, serum creatinine; pos, positive; TMA, thrombotic microangiopathy; U:P/C, urinary protein/creatinine ratio.

[a] extrarenal manifestation of amyloidosis.

[b] at time of diagnosis.

Previous clinical therapy: 1: Bortezomib, 2: Thalidomide, 3: Cyclophosphamide, 4: Dexamethasone, 5: Carfilzomib, 6: Lenalidomid, 7: Daratumumab, 8: Rituximab.

surrounded by a fat pen (Dako) and the staining area was rewetted using PBS. For kappa/lambda staining goat (diluted 1:10000, A0191 and A0193 from Dako) or rabbit anti human kappa/lambda polyclonal antibody (Ab) (P0212, Dako) was applied (1:5000 in PBS) and incubated for two hours at room temperature following two washes in PBS each for 10 min, the donkey anti goat or goat anti rabbit (Alexa 488, dilution 1:1000 in PBS) was applied for 1 h at room temperature. Before washing the slide 20 µL of DAPI solution was spotted onto the Ab containing area. Following the final wash Vetashield mounting media and cover slip were applied and the slides were recorded under a Zeiss invert confocal microscope using ZEN program for picture recording.

## Urinary cast isolation

Urine sediment generated as described above was air dried from patients presenting with AL-amyloidosis and cast nephropathy or presenting with cast nephropathy alone. Cytopreparations were viewed under phase contrast light microscope (Labovert FS, Leitz, Germany) and individual casts were picked using the tip of specifically gas flame pulled glass Pasteur pipettes under microscopic observation. Five casts were placed into a 400 µL PCR tube containing 10 µL PBS. This represented the work up starting material for proteome analysis.

## Substructural fractionation on a sucrose gradient

Cells and cell nuclei and large aggregates were pelleted by centrifugation at 1500 RPM. Five hundred µL of resultant supernatant containing cast fragments and micro-aggregates and exosomes were loaded on to a discontinuous sucrose gradient in Ultra-clear centrifuge tubes (Beckman, 344062) at 4 °C. Following two hours centrifugation at 40000 RPM using an SW60 rotor in an L-80 ultracentrifuge the gradient was immediately fractionated in 200 µL aliquots using a peristaltic pump starting at the bottom of the tube. Individual fractions were subjected to immunoblotting using antibodies for kappa and lambda light chains (Dako as above) and aquaporin 1 (AQP1; Millipore).

## Immunoblotting

Twenty µL of individual fractions of urine were loaded onto a 12% SDS PAGE gel and run under denaturing conditions. Following the transfer onto nitrocellulose the proteins were stained using kappa and lambda specific antibodies (Dako) as well as Ab against AQP1 (Millipore). The specific protein staining was developed by HRP labeled goat anti rabbit affinity purified antibodies (Dako). After each staining procedure the blot was washed using TPBS for 10 min twice. Chemiluminescence reagent was applied to the blot which was visualized using Fusion software at the luminescence recorder (Fusion Fx, Vilber Lourmat). Pictures were further processed using Photoshop version 6. Densitometric evaluation of individual signals of dimers vs. monomers were performed using the Fusion software (Fusion Fx, Vilber Lourmat).

## Uromodulin ELISA test

Uromodulin concentration was measured in the same sucrose density gradient fractions as used for immunoblotting by a commercially available ELISA assay (BioVendor, Brno,

Czech Republic). The Elisa was performed as suggested in the company's test manual. In brief, fractions were diluted in sample dilution buffer and incubated together with the standard series. Following the incubation with biotinylated detection antibody and streptavidin-HRP conjugate the signal was developed with TMB substrate and read by an ELISA reader. Concentrations were calculated according the standard curve. The detection range of the ELISA assay is 0.5 to 25 ng/mL.

## Mass spectrometry
### NanoLC-MS analysis
Trypsin digestion was carried out in solution and resultant peptides were purified on a C18 column before injection into the UltiMate 3000 RSLC nano HPLC system (Thermo Fisher Scientific, Amsterdam, Netherlands). This was connected to a Q Exactive HF mass spectrometer (Thermo Fisher Scientific, Bremen, Germany) with a Proxeon nanospray source (Thermo Fisher Scientific, Odense, Denmark). Initial loading was carried out onto a trap column (Thermo Fisher Scientific, Amsterdam, Netherlands, PepMap C18, five mm × 300 µm ID, 5 µm particles, 100 Å pore size) at a flow rate of 25 µL min-1 using 0.1% TFA as mobile phase. This column was moved in line with the analytical column (Thermo Fisher Scientific, Amsterdam, Netherlands, PepMap C18, 500 mm × 75 µm ID, 2 µm, 100 Å) following a 10 min flow. Peptides were eluted by a binary 1 h gradient with a flow rate of 230 nL min-1. The applied gradient for starting represented: 98% A (water/formic acid, 99.9/0.1, v/v) and 2% B (water/acetonitrile/formic acid, 19.92/80/0.08, v/v/v). Over a period of 60 min B was increased to 35% and over the next 5 min B was further increased up to 90% and was kept in a plateau for 5 min. Within the next 2 min the gradient was returned back to 98% A and 2% B. This was the setting for equilibration at 30 °C.

The mass spectrometer was set to data-dependent mode in full scan (m/z range 380–1,650, nominal resolution of 120,000, target value 3E6) followed by MS/MS scans of the 10 most abundant ions. MS/MS spectra were obtained using normalized collision energy of 27%, isolation width of 2 m/z, resolution of 30,000 and the target value was set to 1E5. Precursor ions selected for fragmentation were read at dynamic exclusion list for 20 s. Additionally, the minimum AGC target was set to 2E4. The intensity threshold was calculated to be 8E4.

### Data Processing protocol and peptide identification
The Proteome Discoverer (version 2.3.0.523; Thermo Scientific) was fed with RAW-data files. The generated MS/MS spectra have been searched using MSAmanda v2.0.0.9849 (*Dorfer et al., 2014*). RAW-files were searched against the SwissProt human database (20,169 sequences; 1,1315,794 residues). The search parameters were: peptide mass tolerance ±5 ppm as well as the fragment mass tolerance was limited to 0.03 Da. Missed cutting sites were 2. The resultant output was filtered to 1% false discovery rate using again Thermo Proteome Discoverer. Peptide area under the curve was quantified using IMP-apQuant (*Doblmann et al., 2019*).

## Statistical analyses

We present quotients as means ± standard deviation (SD). Differences of lambda and kappa dimer vs. monomer ratio were analyzed with paired $t$-test. Data management and analysis was conducted by Microsoft Excel (Microsoft, Redmont, WA, USA). The statistical testing performed was two-sided and a $p$-value $\leq 0.05$ was considered significant.

## Ethics approval and consent to participate

This study involves human participants. Urine samples were obtained from adult study participants and/or their legal guardians older than 18 years who have given written informed consent. All methods of experiments involving human participants were carried out according to relevant guidelines. All experimental protocols involving human participants received approval of the Ethics Committee of the Medical University of Vienna (EK 1043/2016).

## RESULTS

### Cast morphology and constituent analysis

Hematoxylin/Eosin staining of tubular casts revealed various staining intensities as well as colour variation. Colour variation results from different contents, which either are more basophilic or acidophilic depending on constituents. As demonstrated in Figs. 1A, 1C, 1D three casts are more eosinophilic when compared to one in Fig. 1B that shows a fainter staining. The thinner cast at Fig. 1C is originating from a different stretch of the nephron than the other cast fragments, such as the the loop of Henle. Three tubular cells are incorporated in this thin cast.

In order to further study the constituents of tubular casts we sought for the incorporation of light chain components. As already reported in earlier work tubular casts contain the light chain (LC) paraprotein such as also measured in serum. As shown in Figs. 2A and 2B the lambda light chain is incorporated at the tubular cast which has a thinner diameter at the left side upward directed end when compared with the other part. This picture further confirms the assumption that casts can originate from thinner stretches of the nephron.

Analysis of casts by electron microscopy has identified various fine structures in previous studies (*Uribe-Uribe & Herrera, 2006*) and the most dominant variant was represented by the hyaline form, however, the biochemical proteinaceous constituents of hyaline casts in light chain associated kidney disorders has not been the subject of investigation before. Only uromodulin was documented as light chain interaction partner. Therefore we sought by mass spectrometry to analyze the proteome of isolated urinary casts from patient 14 who had no detectable uromodulin in the sucrose density fractionation. In these experiments we identified the patient's lambda LC and histones in particular H2B sub-variants (Table 2) as a dominant additional constituents of the cast proteome. It is of interest that a urinary protease such as cathepsin B was also contained in this complex (Table 2). In repeated proteome analyses ($n = 3$) H2 variants were similarly highly present. Depending on the cast type chosen for analysis more or less cellular proteins were recovered. However, this study was focused on hyaline casts. Of particular note is the observation that as shown in (Fig. S1) patient 14 had received immunomodulatory therapy 2.5 days before the onset

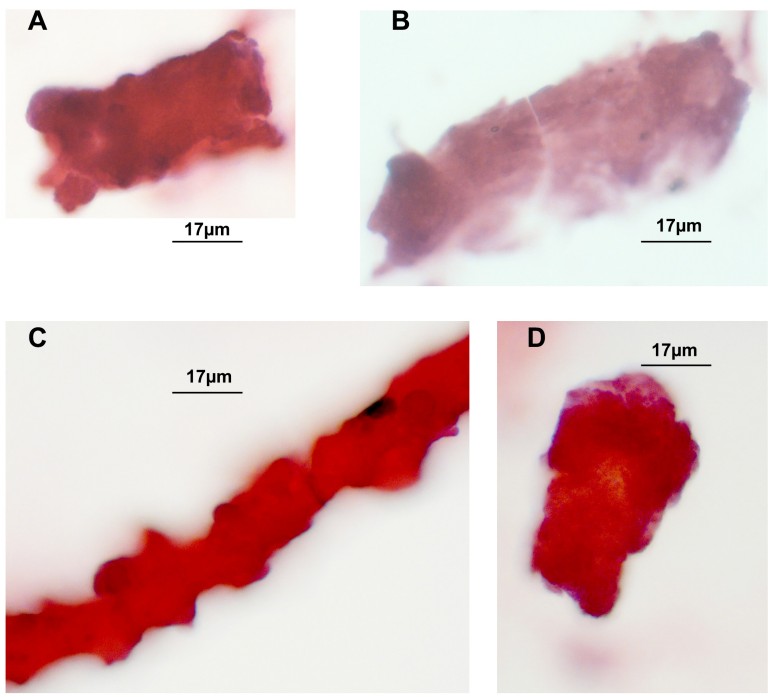

**Figure 1** **H/E stained casts on cyto-slide preparations obtained from a patient with lambda light chain myeloma.** Urine sediment obtained in the context of AKI stage 3 revealed multiple casts. The average cast diameter comprised about 30 µm (A, B, D). Considerable thinner cast containing tubular cells most likely originating from a thinner part of the nephron (C). Pale staining tubular cast originating from the distal tubule (B).

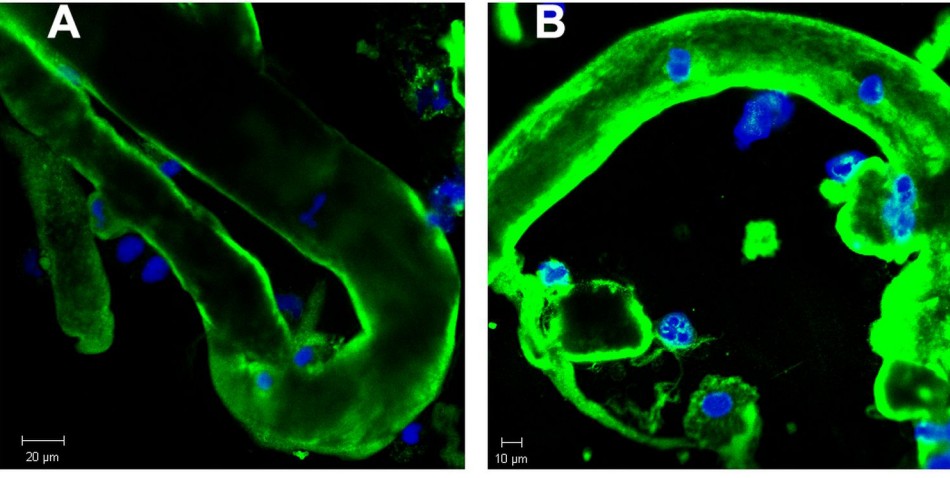

**Figure 2** **Immunofluorescence staining of renal tubular casts. Various staining intensity for lambda light chain of tubular casts.** The cast is bent and shows a conical structure with the thinner part directed upwards. This has a diameter of about 23 µm whereas most other casts show a diameter of about 30 µm (A). Casts contain remnants of tubular cells with condensed nuclear DNA (B).

**Table 2** **Proteome analysis using NanoLC-MS analysis of 5 individual casts selected under microscopic observation by micromanipulation.** Five hyaline casts were picked from a cytoslide and analytically processed. Abundance of peptides is calculated as area under the curve (Norm. Area) of highly specific peptide peaks. Shown is one analysis out of three.

| Accession | Description | MW[kDa] | Norm. Area |
|---|---|---|---|
| P0CG04.1 | Immunoglobulin lambda constant 1 | 11,34 | 3,92E+07 |
| P0DOX8.1 | Immunoglobulin lambda-1 light chain | 22,82 | 2,29E+07 |
| B9A064.2 | Immunoglobulin lambda-like polypeptide 5 | 23,05 | 2,11E+07 |
| P0CF74.1 | Immunoglobulin lambda constant 6 | 11,27 | 1,96E+07 |
| P0DOY3.1 | Immunoglobulin lambda constant 3 | 11,26 | 7,19E+06 |
| P07858.3 | Cathepsin B | 37,80 | 1,49E+06 |
| Q16778.3 | Histone H2B type 2-E | 13,91 | 1,02E+06 |
| Q5QNW6.3 | Histone H2B type 2-F | 13,91 | 1,02E+06 |
| Q99879.3 | Histone H2B type 1-M | 13,98 | 1,02E+06 |
| P33778.2 | Histone H2B type 1-B | 13,94 | 1,02E+06 |
| P58876.2 | Histone H2B type 1-D | 13,93 | 1,02E+06 |
| Q93079.3 | Histone H2B type 1-H | 13,88 | 1,02E+06 |
| Q99880.3 | Histone H2B type 1-L | 13,94 | 1,02E+06 |
| P62807.4 | Histone H2B type 1-C/E/F/G/I | 13,90 | 1,02E+06 |
| Q8N257.3 | Histone H2B type 3-B | 13,90 | 1,02E+06 |
| P23527.3 | Histone H2B type 1-O | 13,90 | 1,02E+06 |
| Q99877.3 | Histone H2B type 1-N | 13,91 | 1,02E+06 |
| P31151.4 | Protein S100-A7 | 11,46 | 9,34E+05 |
| Q96A08.2 | Histone H2B type 1-A | 14,16 | 8,90E+05 |
| O00584.2 | Ribonuclease T2 | 29,46 | 7,65E+05 |
| P01834.2 | Immunoglobulin kappa constant | 11,76 | 6,81E+05 |
| P01860.2 | Immunoglobulin heavy constant gamma 3 | 41,26 | 6,67E+05 |
| Q6ZVX7.1 | F-box only protein 50 | 30,83 | 6,25E+05 |
| Q5T749.1 | Keratinocyte proline-rich protein | 64,09 | 5,34E+05 |
| P0DOX7.1 | Immunoglobulin kappa light chain | 23,36 | 3,40E+05 |
| Q9H1E1.2 | Ribonuclease 7 | 17,41 | 2,86E+05 |
| Q05639.1 | Elongation factor 1-alpha 2 | 50,44 | 1,49E+05 |
| P68104.1 | Elongation factor 1-alpha 1 | 50,11 | 1,49E+05 |
| Q5VTE0.1 | Putative elongation factor 1-alpha-like 3 | 50,15 | 1,49E+05 |
| P00558.3 | Phosphoglycerate kinase 1 | 44,59 | 1,45E+05 |

of AKI stage 3 according to KDIGO. His urine sediment was filled with hyaline casts. The liberation of histones from apoptotic plasma cells and the filtration into the urine might have represented the main provocation of cast formation in the renal tubules as uromodulin was not detectable. His lambda light chain analysis in the density gradient fractions (Fig. 3A) showed to some extent presence of aggregated LC signals in density fractions 6-10 representing most likely complexes but there was almost no presence of LCs in exosome containing fractions 2-6. Presence of exosomes was verified by immunoblotting of the same fractions with Ab against AQP1 a tubular cell membrane protein (Fig. 3B). By a second approach using urine SDS PAGE analysis and immunoblotting lambda LCs

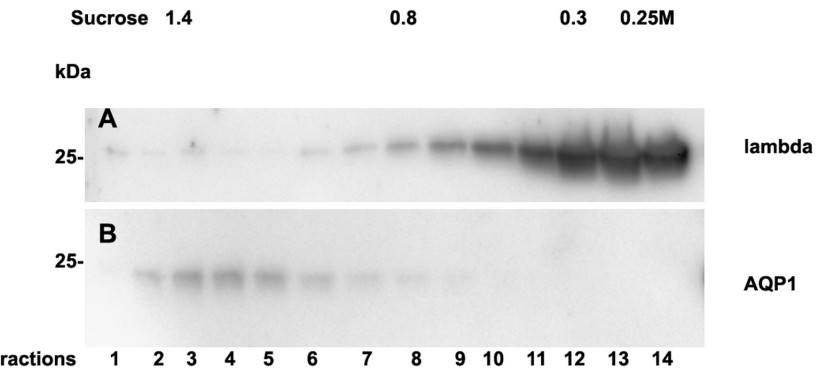

**Figure 3 Sucrose density gradient separation of urine of patient 14 containing cast fragments, exosomes and micro-aggregates.** The centrifuged (40000 g for 2 hours) sucrose gradient (500 μL urine containing cast fragments and micro-aggregates) was fractionated into 14 individual fractions. Individual fractions were loaded onto a 12% SDS PAGE gel, transferred to nitrocellulose and developed for lambda light chains (A). The same fractions were reprocessed and blotted for aquaporin 1 (AQP1) as a marker protein for heavy exosomes entering the dense sucrose layers such as in fractions 2–6 (B).

appeared predominantly as dimers in urine and made up 78% in the densitometric analysis (Fig. 4, lane 14). No amyloid deposition had been observed in the kidney biopsy of this patient although the duration of disease had been lasting for already 5 years (Table 1). The patient underwent high cut-off dialysis and as demonstrated in Fig. S1 his renal function improved.

## Analysis of urinary light chain structure

As already documented in earlier work not the concentration but rather the structure and morphology represents the underlying cause for manifestation of renal light chain disease. For this reason we analyzed the light chain dimerization in 17 myeloma or monoclonal gammopathy patients (Table 1). As demonstrated in Fig. 4 patients positive for lambda LCs showed patterns with higher quantities of LC dimers than monomers. Furthermore, eight patients out of 11 positive for lambda LC developed CKD stages 4–5 according to KDIGO (Table 1). Patients 3, 8, 9, 10, 13 and 14 were highly positive for lambda dimers in the immunoblots (Fig. 4). Dimerization of lambda LC could be entirely resolved into monomers by adding DDT as reducing agent in the sample buffer (Fig. 4B).

Patients positive for kappa LCs showed patterns with higher quantities of LC monomers than dimers (Fig. 5). The densitometric analysis evaluated ratio of lambda dimers vs. monomers was significantly higher (2.12 ± 0.75) when compared with the ratio of kappa dimers vs. monomers (0.64 ± 0.47), $p = 0.00001$.

We further intended to elaborate the light chain aggregates or complex formation resulting in renal light chain disease by density gradient centrifugation in patients 13, 14 and 15. In the urine sediment of amyloidosis patient 13 the light chain was found in both the soluble urine proteins (Fig. 6A, fractions 11–14) and to a lesser extent in the membrane fractions (Fig. 6A, lanes 1–4). Membrane exosome fractions were identified by AQP1 blotting of the same fractions (Fig. 6B). This was similar for patient 15 despite the fact

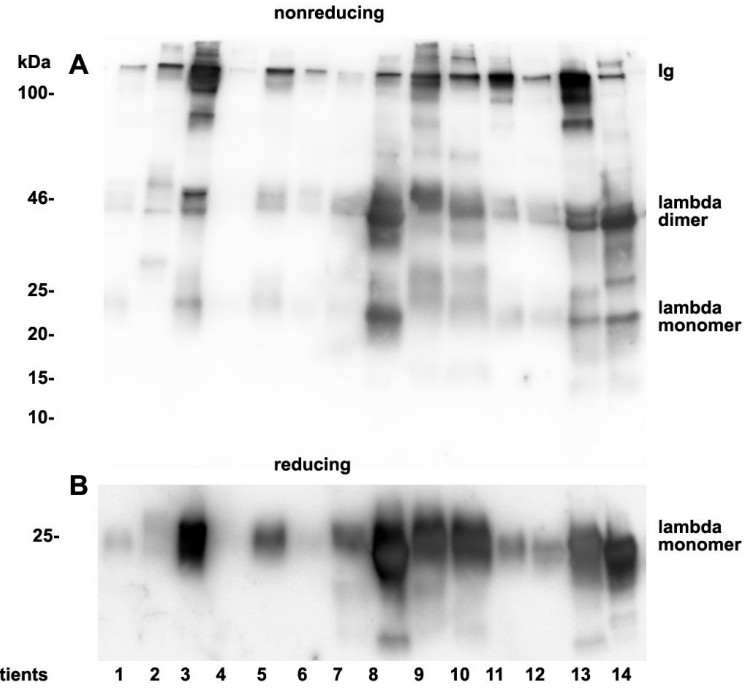

**Figure 4 Urine lambda LC immunoblotting.** Twenty μL urine were loaded onto a 12% SDS PAGE gel and run under non-reducing conditions. The gel was blotted onto nitrocellulose and developed using a lambda LC-specific Ab (A). The clinical course of the patients and predominant LCs are given in Table 1. The same fractions were loaded onto a 12% SDS PAGE gel and run under denaturing and reducing conditions. The gel was blotted onto nitrocellulose which was developed with the lambda LC-specific Ab. Only monomeric forms of LCs were depicted (B). Patients are numbered at the bottom of the picture in accordance to Table 1. The molecular weight is shown on the left side and light chain multimerization status is shown on the right. Ig stands for immunoglobulin.

that his disease was linked to kappa LC which was going along with AKI stage 3 (Fig. 7A). Membrane fractions represented by lanes 1–4 containing exosomes and cast particles were identified by presence of AQP1 (Fig. 7B).

The Uromodulin protein (UMOD) has been shown earlier to represent the binding partner for LCs in forming casts. In this study we did not find uromodulin protein in the casts of patient 14, which was confirmed by absence in sucrose gradient density fractions and urine. In contrast, there was a small peak concentration in heavy membrane fraction 2 in patients 13 and 15 (Figs. 6C and 7C). This fractions corresponded with presence of exosomes and cast fragments in the gradient sedimentation experiments. Fractions 11–14 correspond to soluble molecules in urine. In patients 13 and 15 urine uromodulin concentration was high (Figs. 6C and 7C).

## DISCUSSION

This study investigates hyaline cast constituents and factors involved in cast formation as well as urine light chain multimeric conformation. It documents that casts of thinner diameter than assumed can be observed in light chain disease which might originate from

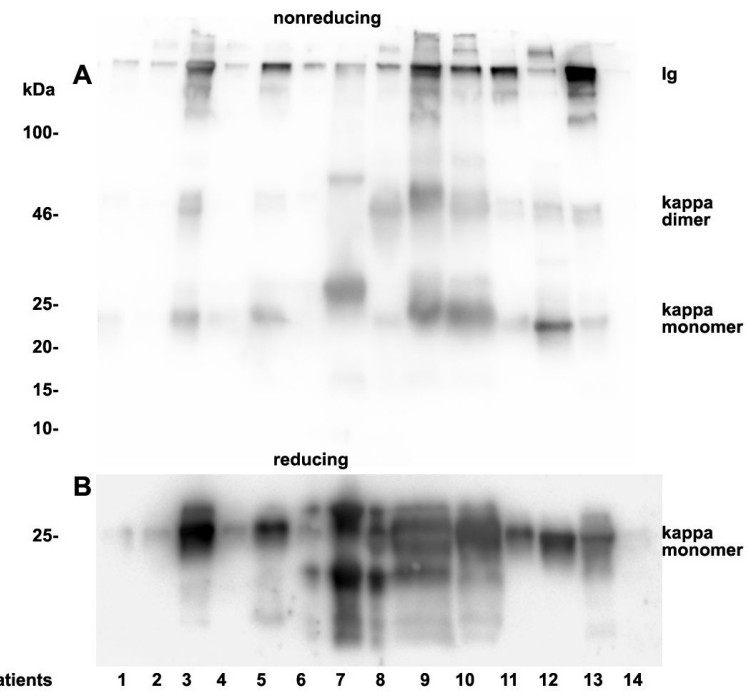

**Figure 5  Urine kappa LC immunoblotting.** Twenty µL urine were loaded onto a 12% SDS PAGE gel and run under non-reducing conditions. The gel was blotted onto nitrocellulose and developed using a kappa LC-specific Ab (A). The patients' clinical conditions and predominant LC are illustrated in Table 1. The same fractions were loaded onto a 12% SDS PAGE gel and run under reducing and denaturing conditions. The gel was blotted onto nitrocellulose which was developed with the kappa LC-specific Ab. Only monomeric forms of LCs were depicted (B). Patients are numbered at the bottom of the picture in accordance to Table 1. The molecular weight is shown on the left side and light chain multimerization status is shown on the right. Ig stands for immunoglobulin.

the loop of Henle. Selective cast harvesting by micromanipulation under microscope observation for proteome analysis revealed histones as major constituents of casts together with the light chain present as serum and urine paraprotein. Rather unexpectedly, cathepsin B was also found among the dominant cast constituents. In addition, all patients with lambda light chains showed dimerization of urine light chains in the immunoblot and were more likely to develop amyloidosis and chronic kidney disease.

Secretion of light chains has been shown in previous studies to predominantly occur as monomers *in vitro* (*Dul et al., 1996*) but this does not have to apply to multiple myeloma patients *in vivo*. In our study patients positive for lambda LCs showed patterns with higher quantities of LC dimers than monomers in urine immunoblotting. In contrast, patients positive for kappa LCs showed patterns with higher quantities of LC monomers than dimers. This may represent an important observation for potential clinical treatment options such as high cut-off dialysis (*Bridoux et al., 2017*; *Finkel, 2014*). The much smaller monomeric forms might represent better candidates for clearance through dialysis, whereas dimers may require longer dialysis and may be less cleared from plasma. It would therefore

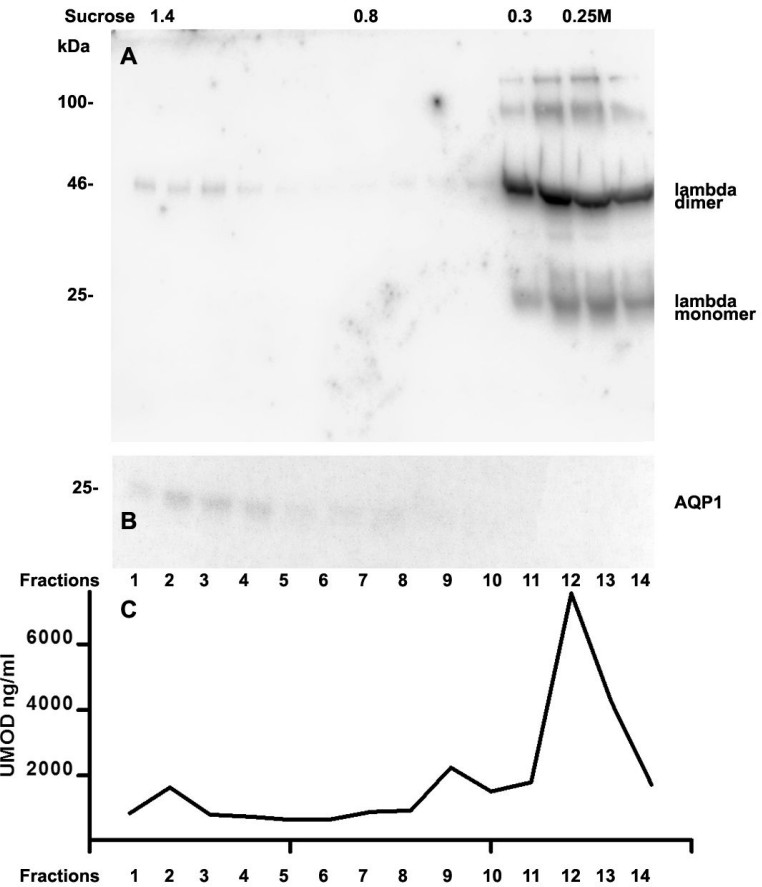

**Figure 6** **Sucrose density gradient separation of 500 μL urine of patient 13 containing cast fragments, micro-aggregates and exosomes.** The centrifuged (40,000 g for 2 hours) sucrose gradient was fractionated into 14 individual fractions which were loaded onto a 12% SDS PAGE gel, transferred to nitrocellulose and developed for lambda light chains. Molecular weight is depicted on the left, lambda dimers and lambda monomers are indicated at the right, fraction numbering is indicated at the bottom and sucrose density at the top (A). The same fractions were tested for AQP1 (B) and for uromodulin concentration by ELISA (C).

be reasonable to test patients for LC structure and size morphology to assess the potential benefit of high cut-off hemodialysis.

Earlier work (*Korbet & Schwartz, 2006*; *Luque et al., 2017*) and this study confirm that cast formation and amyloid fibrilogenesis (*Kim et al., 2000*) represent complex physicochemical and biochemical processes (*Radamaker et al., 2019*). Interaction of electrostatic charges between proteins such as uromodulin (*Huang & Sanders, 1997*) together with nephron-specific biological processes such as incorporation of exosomes shaded from tubular epithelial brush border are of importance. In addition, proteins filtered through the glomerulum and not reabsorbed by the proximal tubular epithelia seem to represent important players in hyaline cast formation.

In our cast experiments histones getting liberated from apoptotic cells during chemotherapy, representing small positively charged molecules of 13–17 kDa, which

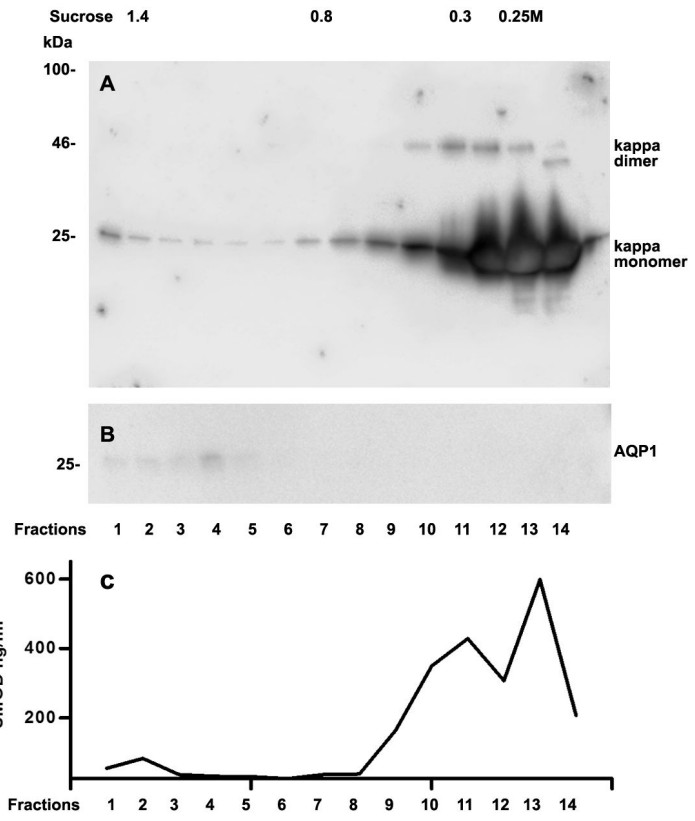

**Figure 7 Sucrose density gradient separation of urine containing cast fragments, micro-aggregates and exosomes from patient 15.** The centrifuged (40,000 g for 2 h) sucrose gradient (500 μL urine containing cast fragments and micro-aggregates) was fractionated into 14 individual fractions. Individual fractions were loaded onto a 12% SDS PAGE gel, transferred to nitrocellulose and developed for kappa light chains. The legend to kappa monomers and dimers is indicated on the right (A). The same fractions were tested for AQP1 (B) and for UMOD by ELISA (C).

are filtered through the glomerular membrane, seem to have acted as partners for LCs to form hyaline casts. In this situation no uromodulin concentrations in the urine could be measured of the patient having undergone immunomodulatory therapy and AKI. Thus it has to be suggested, that cast formation might be induced by various proteins and biological compounds (*Luque et al., 2017*) beyond uromodulin. Hence, novel considerations in management of cast nephropathy might be important (*Manohar, Nasr & Leung, 2018*; *Ying et al., 2012*).

It has been documented earlier that Bence Jones protein and amyloid fibril generated out of LCs are C-terminally cleaved (*Enqvist, Sletten & Westermark, 2009*; *Terry et al., 1973*). In this line the abundant presence of Cathepsin B in hyaline casts represents a protease known for cleaving C-terminal parts in the process of activation/inactivation of various biologically active compounds (*Kumar et al., 2018*). It is therefore of interest that Cathepsin B might be involved in proteolytic processing of the C-terminal part of the cast components in particular the LCs. Cathepsin B is most likely involved in proteolytic digestion and thereby mobilization of casts from the site of tubular obstruction. It is of note that in fraction 14 of

Fig. 7A (soluble urine proteins) the light chain dimer is of reduced size probably because of cleavage, which must have been achieved by urine contained proteases such as cathepsins.

Brush border specific proteins are contained in the hyaline casts isolated from the urine sediment in individual specimens. Some of the patient derived hyaline casts contained cells to various extent. This demonstrates that cast formation at a specific stretch of the nephron causes injury to tubular epithelial cells or cells subsequently loosen from the basal membrane when the tubule dilates proximal to the part, which had been obstructed by the cast. Proteases incorporated (such as cathepsin) might contribute to cast liberation because of proteolytic digestion and thereby shrinkage of the cast.

We found kappa and lambda LCs in the heavy fractions of the sucrose gradient at densities where exosomes and cast fragments were expected. This revealed that LCs are present in microsomes where AQP1 is present or it represents cast fragments.

The molecular size of LC in serum must have an important impact on its clearance by the glomerulum and by high cut-off dialysis membranes. LC testing using urine immunoblotting to determine the molecular size of LC proteins appears to be an important option for evaluating whether a patient could benefit from high cut-off dialysis treatment. Lambda-positive patients known for the presence of higher polymers (*Sallee & Burtey, 2019*), as confirmed in this study, may not be good candidates for LC clearance through high-cut off dialysis membranes.

The main limitation of this study is that no clinically available quantitative methods are available for measurement of LC dimers versus monomers, therefore this study uses semi-quantitative methods. Casts for proteomic analysis could not be isolated from all patients and not all patients underwent renal biopsy. Due to the limits of proteomic databases, only well-known peptide sequences can be detected by proteome analysis. In respect of clinical treatment regimens this structural analysis of casts and light chains is preliminary and focused on a small sample size. This may encourage clinical researchers at the same time to conduct studies in larger patient groups on different treatment regimens, in particular on lambda/kappa light-chain associated kidney disorders. It is of note that, in our study for the first time, proteomic analysis was performed on micro-selected cast structures.

## CONCLUSIONS

This study demonstrates that hyaline tubular casts represent a complex formation of protein-protein aggregates with histones and cathepsin B identified as novel cast components. Apart from the proteomic cast composition, the formation of light chains and aggregates is of particular significance. Dimerized light chains typical for lambda paraproteins might be less dialyzable than monomeric forms. Therefore, testing for light chain structure may select patients who are more likely to benefit from dialysis using high cut-off membranes, but clinical trials are required to confirm this.

### Funding
The authors received no funding for this work.

### Competing Interests
The authors declare there are no competing interests.

### Author Contributions
- Thomas Reiter performed the experiments, analyzed the data, prepared figures and/or tables, approved the final draft.
- Daniela Knafl prepared figures and/or tables, approved the final draft.
- Hermine Agis contributed reagents/materials/analysis tools, approved the final draft.
- Karl Mechtler conceived and designed the experiments, performed the experiments, approved the final draft.
- Ludwig Wagner conceived and designed the experiments, performed the experiments, analyzed the data, contributed reagents/materials/analysis tools, authored or reviewed drafts of the paper, approved the final draft.
- Wolfgang Winnicki conceived and designed the experiments, prepared figures and/or tables, authored or reviewed drafts of the paper, approved the final draft.

### Human Ethics
The following information was supplied relating to ethical approvals (i.e., approving body and any reference numbers):

The Ethics Committee of the Medical University of Vienna granted Ethical approval to carry out the study within its facilities (Ethical Application Ref: 1043/2016).

### Data Availability
Raw data are available in Figures 1-7, Table 1 and the Supplemental Files.

### Supplemental Information
Supplemental information for this article can be found online at http://dx.doi.org/10.7717/peerj.7819#supplemental-information.

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
