# Peer review of "Structural analysis of urinary light chains and proteomic analysis of hyaline tubular casts in light chain associated kidney disorders"

_PeerJ, doi:10.7717/peerj.7819_

## Round 0.1 · original submission · Minor Revisions

Dear Dr. Winnicki,

Your manuscript entitled " Structural analysis of urinary light chain architecture and proteomic analysis of hyaline tubular casts in renal light chain disease" which you submitted to PeerJ, has been reviewed by the editor and 2 experts in the field.

Both reviewers found the paper to be clear and the findings to be of interest. However, they raised some concerns which need to be considered. If these can be satisfactorily addressed, a revised manuscript is likely to be suitable for publication. I enclose below the comments received that set out a number of points which will need your attention before we can consider the submission further. I would urge you to give these points your careful attention; in particular, I encourage you to address the comment about technical aspects of the analysis, patient classification and description of the clinical features (e.g., renal pathological findings and hematological characteristics) of the patients. A section that addresses the study's limitations is also required; technical/protocol details, potential limitations of experimental design etc., must be openly discussed.

I hope that you will be prepared to make the necessary amendments and submit a revised manuscript accompanied by a statement of how you have responded to the criticisms raised. Please copy and paste each and every reviewer's comment above your response. If you feel any of their points are inappropriate, you are certainly free to provide rebuttal in your covering letter.

Please note that resubmitting your manuscript does not guarantee eventual acceptance. I must emphasize that the acceptability of the revision will depend upon the resolution of the points raised by the reviewers.

Sincerely yours,

Stefano Menini

·

Basic reporting

The paper is clear and well written. The language used is technically appropriate. Both the background and the objectives are clearly described in the introduction as in the discussion. The authors demonstrate to deeply know the previous literature in the specific field of investigation.
The table and figures are clear, well detailed and labeled. Legends are exhaustive. Although the absence of uromodulin and the consequent deductions on the formation of the casts in the considered patient appear relevant (lines 233-246 in the text), the Figure 3C should be removed and the result of the related experiment should be reported just in the text.
For the data shown in figure 4, it would be advisable to be reported as a table, maybe using the format proposed by the software used to analyze (and appropriately filter) the large amount of data got by proteomic experiments.

Experimental design

The research questions are well defined and discussed throughout the text. The experiments are rigorous, taking into account the technological approaches used today for such these studies.
Although this is a preliminary study, to move to clinical trials the statistics of some data and the strengthness of the conclusions should be improved increasing the total number of patients involved. In particular, data relating to a single patient, or to a few patients, should be verified on similar population of patients, even considering, as was correctly done in this work, the pharmacological treatments to which they have been subjected.
The technical aspects of proteomic nano LC-MS analysis are partial and, although they refer to previous studies, they cannot fail to report some essential technical information to allow these experiments to be reproduced by other researchers. In particular, the proteases used to obtain the peptides, subsequently analyzed and characterized, as well as the digestion procedure (in solution, in column; why not in gel?, etc.), should be described in the detail.

Validity of the findings

The results are novel and partly confirm what has been shown in previous studies. This study undoubtedly contains interesting data for the advancement of knowledge in the study and treatment of patients with light chain disease. Although circumscribed to a limited and, in some respects, heterogeneous number patients the obtained data can be meaningful to think of developments that lead to an optimized, single patient, dialysis treatment.

·

Basic reporting

The authors studied urinary casts composition in 4 patients with light chain associated kidney disorders. The method seems robust. However, detailed renal biopsy findings in these patients are lacking. The second part of this study focus on urine light chain examined by immunoblotting in 17 patients with monoclonal gammopathy.
1. Would consider revising the title. Perhaps the word “architecture” could be deleted and maybe “light chain associated kidney disorders” instead of "renal light chain disease”.
2. The term “light chain nephropathy” (for example line 53) is not appropriate and may be confusing. The authors should prefer “light chain associated kidney disorders”.
3. Since multiple myeloma is a plasma cell dyscrasia, the distinction used (for example line 52) is not appropriate. The authors should classified patients with symptomatic multiple myeloma or monoclonal gammopathy of renal significance (MGRS) since this distinction is more relevant in the context of paraprotein associated kidney disorders.
4. Your introduction should better highlight that the spectrum of renal complications associated with paraproteins is heterogeneous with two main categories according to the characteristics of the underlying B-cell clone. The first group is always associated with high tumor mass and light chain cast nephropathy is the more frequent complication in this context. The second group include all patients with renal symptoms related to a nephrotoxic monoclonal immunoglobulin by any mechanism other than the tumor burden and is now defined as MGRS (ref Leung N et al. Blood 2012).
5. The reference Leung N et al. Nature reviews Nephrology. 2019 (https://doi.org/10.1038/s41581-018-0077-4) needs to be added to the introduction section.
6. Line 64 : free light chain measurement in the urine is not recommended in the management of patients with paraprotein associated kidney disorders. Please discuss this point.
7. Line 79 : please replace “renal deposition disease” by “monoclonal immunoglobulin deposition disease”

Experimental design

No comment

Validity of the findings

Renal biopsy was not performed in all patients. This point is an important limitation of the study. How many patients had a renal biopsy? What means “0” in column renal pathology in Table 1?

Additional comments

The authors should better described renal pathological findings and hematological characteristics of the patients included in this study. You should distinguish in Table 1 patients with symptomatic multiple myeloma or MGRS.

Reviewer 3 ·

Basic reporting

Experimental design

Validity of the findings

Additional comments

"Structural analysis of urinary light chain architecture and proteomic analysis of hyaline tubular casts in renal light chain disease"

The work is well conducted, the main topic is clear and the description of the procedure is really well done. The laboratory part is very accurate and all the conclusions are strictly clear and give messages that can be really useful for clinicians and for researchers.

Overall is a very good article

The only remark is about clinical aspects:

* authors describes only architectural and structural composition of casts, basing the whole work on the morphological carachteristic of chains. Nevertheless, they don't describe number of free chains, creatinine at the time of diagnosis, previous clinical treatment patients underwent (a number of patients has a long history of mieloma) making difficult to evaluate the patient in all his complexity. Lots of datas about the kind of free chains and casts could depend on the patients previous conditions and it could be useful to describe.
* giving the limited number of patients seems difficult to divide into subgroups but the age of patients and the age of illness could play an important role in histones precipitation and reaggregation.
* at the beginning authors describes only three forms of renal involvment during multiple myeloma while there are a few more (Immunotactoid, crioglobulinemic nephrotic syndrome, leukaemia invasion, nephropaties with organised Immunoglobulin deposits). The description of all the different kind of lesions is very important in the context of electronic microscopy.

---

## Round 0.2 · accepted · Accept

Dear Drs. Wagner and Winnicki,

Our referees have now considered your manuscript entitled "Structural analysis of urinary light chain architecture and proteomic analysis of hyaline tubular casts in renal light chain disease" and have recommended publication in “PeerJ”. We are pleased to accept your paper in its current form which will now be forwarded to the publisher for copy editing and typesetting.

I thank all reviewers for their effort in improving the manuscript and the authors for their cooperation throughout the review process

Yours sincerely,

Stefano Menini

·

Basic reporting

Since the authors have implemented all the suggested changes , in my opinion the manuscript can be accepted for publication

Experimental design

Since the authors have implemented all the suggested changes , in my opinion the manuscript can be accepted for publication

Validity of the findings

Since the authors have implemented all the suggested changes , in my opinion the manuscript can be accepted for publication

·

Basic reporting

No comment

Experimental design

No comment

Validity of the findings

No comment

Additional comments

No comment